



# Scant evidence for a volcanically forced winter warming over Eurasia following the Krakatau eruption of August 1883

Lorenzo M. Polvani[1,2,3] and Suzana J. Camargo[2,3]

[1]Department of Applied Physics and Applied Mathematics, Columbia University, New York, NY 10027, USA
[2]Department of Earth and Environmental Sciences,Columbia University, New York, NY 10027, USA
[3]Lamont-Doherty Earth Observatory, Columbia University, Palisades, NY 10964, USA

**Correspondence:** Lorenzo M. Polvani (LMP@COLUMBIA.EDU)

**Abstract.** A recent study has presented compelling new evidence suggesting that the observed Eurasian warming in the winter following the 1992 Pinatubo eruption was, in all likelihood, unrelated to the presence of volcanic aerosols in the stratosphere. Building on that study, we here turn our attention to the only other low-latitude eruption in the instrumental period with a comparably large Volcanic Explosivity Index (VEI): the Krakatau eruption of August 1883. We study in detail the temperature

anomalies in the first winter following that eruption, analyzing (1) observations, (2) reanalyses, and (3) models. Three findings emerge from our analysis. First, the observed post-Krakatau winter warming over Eurasia was unremarkable (only between 1- and 2-$\sigma$ of the distribution from 1850 to present). Second, reanalyses indicate the existence of very large uncertainties, so much so that a Eurasian cooling is not incompatible with observations. Third, models robustly show the complete absence of a volcanically forced Eurasian winter warming: we here analyze both a 100-member initial-condition ensemble, and 140

simulations from the Phase 5 of Coupled Model Intercomparison Project. This wealth of evidence strongly suggests that, as in the case of Pinatubo, the observed warming over Eurasia in the winter of 1883/84 was, in all likelihood, unrelated to the Krakatau eruption. Together with the results for Pinatubo, we are led to conclude that if volcanically forced Eurasian winter warming exists at all, an eruption with a magnitude far exceeding these two (VEI=6) events is needed.

## 1  Introduction

The vexed question of whether large, low-latitude volcanic eruptions are able to cause a winter warming of the continents in the Northern Hemisphere is receiving renewed attention. As the extant literature is quite confusing and often contradictory, a brief summary of how this topic has evolved will help set the stage for the present study.

The surprising idea of a post-eruption *warming*, obviously at odds with the naive expectation of a surface cooling from addi-

tional reflection of incoming short-wave radiation by the volcanic aerosols, was championed by a series of early observational studies (Groisman, 1992; Robock and Mao, 1992, 1995). These were accompanied by modeling studies (Graf et al., 1993; Kirchner et al., 1999) which reported a causal relationship between strong eruptions and a positive phase of the North Atlantic





Oscillation (NAO) in winter. These early studies were highly influential because they proposed a cogent physical mechanism linking low-latitude eruptions to high-latitude surface temperatures anomalies. In a nutshell: the lower stratospheric tropical

warming associated with the absorption of long-wave radiation by the volcanic aerosols increases the equator-to-pole strato-spheric temperature gradients, resulting in an anomalously strong polar vortex leading, in turn, to a positive NAO phase, and eventually a surface warming over Eurasia in the winter months (Kodera, 1994; Perlwitz and Graf, 1995). We will refer to this mechanism as the "stratospheric pathway".

    While such a mechanism appears plausible, these early observational and modeling claims have not stood the test of time.

For instance, analyzing 15 eruptions in one temperature reconstruction extending back to 1586, Fischer et al. (2007) found that the largest winter warming over Europe seems to occur in the second post-eruption winter, not the first. If confirmed by other reconstructions, this result would obviously negate the stratospheric pathway mechanism, as very little aerosol is left in stratosphere in the second post-eruption winter, and no physical process has been proposed that would provide such a long memory in the polar stratosphere and the NAO. Even more perplexing are the numerous studies (Thomas et al., 2009; Driscoll

et al., 2012; Toohey et al., 2014; Bittner, 2015; Wunderlich and Mitchell, 2017, just to cite the most recent ones), which have analyzed, literally, dozens of state-of-the art climate remodels and repeatedly failed to replicate the early modeling results. Averaging over many different low-latitude eruptions, over many models, and over many runs, these more recent studies have consistently reported the lack of a statistically significant post-eruption surface winter warming over Eurasia. This begs the question: how could the later, better models fail to simulate the causal connection reported with the earlier, simpler models?

A resolution to this conundrum was recently proposed by Polvani et al. (2019), hereafter PBS19, who showed that much of the earlier literature had failed to properly account for the large internal variability associated with the stratospheric polar vortex and with the NAO. Focusing primarily on the 199 Pinatubo eruption, PBS19 analyzed three "large ensembles" of model runs, and showed how tiny initial perturbations of the *same* model forced with the *same* volcanic aerosols result vastly different winter temperature anomalies over Eurasia. They also showed that, averaging over many runs, i.e. computing the "forced"

response, yields insignificant post-eruption anomalies, in agreement with most recent studies. The early modeling studies, it so happens, simply lacked the vertical and horizontal resolution to properly simulate the internal variability of the Northern Hemisphere circulation, and thus yielded spurious results. PBS19, therefore, concluded that the observed Eurasian surface warming in the winter of 1991/1992 was, very likely, *not* caused by the preceding eruption of Mt Pinatubo.

    That conclusion appears unassailable, since PBS19 merely corroborated the lack of a forced post-eruption winter warming

already reported by many previous studies with many other models. Nonetheless, one might argue that the 1991 Pinatubo eruption was somehow peculiar, and thus unrepresentative of most, large, low-latitude eruptions. To determine if this is so different eruptions need to be examined. PBS19 did, in fact, examine the 1983 El Chichón eruption (Robock, 1983) with the same three large ensembles, and again concluded that the winter warming over Eurasia in the winter 1982/83 was, in all likelihood, *unrelated* to El Chichón. But this is perhaps unsurprising, as the El Chichón eruption was of smaller magnitude what

the Pinatubo eruption. What is really needed, it should be clear, is a low-latitude eruption at least comparable in magnitude, if not larger, than the last Pinatubo eruption.



Hence this study: building on PBS19, we here examine the eruption of Mt. Krakatau in August of 1883. There are several reasons for focusing on that eruption. First, it is the only low-latitude eruption that occurred during the instrumental period with a magnitude comparable to the 1991 Pinatubo eruption (Robock, 2000), in both volcanic explosivity index (VEI) and dust veil

index (DVI=1000): this means that we can establish, with some confidence, whether Eurasia was anomalously warm or cold in the first winter after the eruption. Second, the Twentieth Century Reanalysis (Compo et al., 2006) starts in 1851, and thus includes that eruption: in fact, not only single renalaysis, but a 56-member ensemble of that reanalysis is now available, so that the observational uncertainties can be quantified. Third, the historical integrations of dozens of models participating in Phase 5 of the Coupled Model Intercomparison Project (hereafter CMIP5; Taylor et al., 2012) also start at 1850: there is, therefore, a

wealth of model output that covers that eruption, and we will examine both a large initial condition ensemble (100 members), and the multi-model CMIP5 ensemble.

As we will show in detail below, this wealth of data allows us to establish the three facts: 1) that Eurasia was indeed warmer than average in the winter following the 1993 eruption of Mt. Krakatau, but 2) that the anomalous warming that winter was in no way exceptional, and 3) that it is unlikely that volcanic aerosols were the *cause* of, or even substantially contributed to, that

warming. After a brief section on methods, each of these facts will be discussed in detail in Sections 3-5. The paper will close with a summary and discussion.

## 2    Methods

### 2.1    Definitions

For each of the datasets detailed in the following subsection, we quantify the potential impact of the 1883 Krakatau eruption

by computing a *winter temperature anomaly*, defined as the difference between the surface temperature in the winter 1883/84 (i.e the average of December 1883, January 1884, and February 1884) and the mean over a reference period, chosen to be five previous winters (i.e. from 1978/89 to 1882/1883). For brevity we will sometime refer to this quantity as "the post-Krakatau anomaly", or simply "the anomaly." Unless otherwise stated, the length of five seasons is chosen for the reference period throughout this paper; this is done to remain consistent with PBS19, so that the post-Krakatau anomalies may be quantitatively[1]

compared with the post-Pinatubo anomalies. In the figures, we will use $\Delta T_s$ to refer to these temperature anomalies.

Furthermore, as explained in details in PBS19, and also suggested by Zambri and Robock (2016) and Bittner et al. (2016a), there is no plausible reason to consider the second winter after the eruption, as no study to date has proposed a credible mechanism that would allow anomalies caused by the volcanic aerosols to affect the stratospheric polar vortex and the NAO beyond the first winter. Finally, to compute time series and probability distribution functions, the average Eurasian anomaly

---

[1]Stenchikov et al. (2006), Driscoll et al. (2012), Bittner et al. (2016a) and most other studies, opted to use reference periods of different lengths for different eruptions. We understand their motivation for doing so: compositing the longest possible number of unperturbed years prior to each eruption. Our goal, however, is a little different: we are trying to validate the Pinatubo findings of PBS19 with a different eruption. Hence, we prefer to use the same reference period as theirs, to avoid introducing a source of potential confusion. In any case; we shown below that the results are largely insensitive to the length of the reference period.





is computed as the mean over the region (40-70°N, 0-15°0W), as in PBS19. We will refer to this as the "Eurasian box." Our results are insensitive to this choice, as one can see from the actual anomaly maps, shown throughout the paper.

## 2.2 Datasets

To construct a comprehensive and convincing picture of potential impact of the 1883 Krakatau eruption on wintertime Eurasian temperatures, we here analyze three different types of surface temperature data: observational reconstructions, reanalyses and 90 model output. We describe each one in detail in the following subsections.

### 2.2.1 Observations

Three distinct datasets which have constructed observational estimates of monthly-averaged temperatures are here analyzed:

1. The NOAA merged Global Surface Temperature (NOAAGlobalTemp) dataset, version 4.00, spanning the period January 1880 to present (Smith et al., 2008; Vose et al., 2012). We will refer to this as the NOAA anomalies.

2. The GISS Surface temperature analysis, version 4, spanning the period January 1880 to present (Lenssen et al., 2019). We will refer to this as the GISTEMP anomalies.

3. The Climate Research Unit (CRU) air temperature anomalies, version 4, spanning the period January 1850 to present (Jones et al., 2012). We will refer to this as the CRUTEM anomalies.

While all three cover the 1883 Krakatau eruption, both the NOAA and GISTEMP products start at 1880, so only three 100 winters are available to define a pre-eruption reference period. We test the robustness of the estimated anomalies as a function of the reference period in Section 3 below.

### 2.2.2 Reanalyses

At the time of this writing, the majority of available reanalyses reach back to 1979, the start of satellite era, or a little earlier (e.g. JRA-55, Kobayashi et al., 2015, which starts at 1958), with only a few extending back to the beginning of the 20th century (e.g. 105 ERA-20C, Poli et al., 2016): unfortunately, none of these serve our purposes. However, the NOAA-CIRES Twenty Century Reanalysis, Version 2c (hereafter 20CR, Compo et al., 2011) extends back to January 1851, and can thus be used to study the 1883 Krakatau eruption. Recall that 20CR only assimilates sea level pressure measurements (Compo et al., 2006), while the underlying atmospheric model is forced with SST reconstructions (Rayner et al., 2003) and time varying $CO_2$ concentrations, incoming solar radiation and volcanic aerosols. More importantly, we here analyze a 56-member ensemble of 20CR: this allows 110 us to quantify the uncertainties associated with the post-eruption Eurasian temperature anomalies.

### 2.2.3 Models

It so happens that a huge amount of model output is available to study the 1882 Krakatau eruption. To a large degree, this is due to the fact that the CMIP5 project recommended that all so-called "historical" integrations should be initialized at the year




1850. Over that period, the CMIP5 also specified how models ought to be forced, with both natural (i.e. solar and volcanic) and anthropogenic (e.g. $CO_2$) forcings, in order be directly comparable with observations. Specifically, we here analyze two distinct datasets.

1. First, we consider the 100-member of historical simulations of so-called "MPI Grand Ensemble" produced by the Max Planck Institute for Meteorology (hereafter MPI, Maher et al., 2019). The atmospheric component of that model (MPI-ESM-LR) has a horizontal resolution of 1.9° (approximately, at the equator), and 47 vertical levels with a model top at 0.01 hPa. As such it is a so-called "high-top" model, with a good representation of the stratospheric circulation and, more importantly, of its variability. We note, in passing, that the other widely used large ensemble (Kay et al., 2015) was initialized at the year 1920, and therefore cannot be used for the purposes of this study. All simulations in this ensemble are performed with an identical model configuration and with identical forcings, following the CMIP5 protocol: they only differ in their initial conditions. This ensemble of runs allows us to unambiguously quantify the importance of internal variability, and contrast it with the forced response.

2. Next, we consider the CMIP5 historical (1850-2005) integrations: the output of 40 distinct models is available, with many groups contributing small ensembles of runs (typically 3 to 10 members), for a total of 140 historical runs. The specific models analyzed here, and the number of integrations available, are listed in Table 1. We examining all available simulations from these models, sorting the models into two categories: high-top and low-top, following the classification in Charlton-Perez et al. (2013) and Rea et al. (2018). If, as argued in the early studies, the stratospheric pathway is indeed operative, one should be able to see a clear difference between the models with a good representation of the stratosphere and those with a poorer representation.

## 3 Observations

We start by determining whether, in the first winter after the Krakatau eruption of August 1993, Eurasia was anomalously warm or anomalously cold. It is important to stress that for most eruptions occurring before the 19th century, not only is the strength of the eruption difficult to estimate, but even the surface temperature anomalies over Eurasia are only known with considerable uncertainties, as those eruptions predate the instrumental period. Hence the 1883 Krakatau eruption offers the only opportunity – other than the 1991 Pinatubo eruption – to examine a large, low-latitude event for which the surface temperature anomalies can be determined robustly.

The post-Krakatau wintertime anomalies, for the CRUTEM dataset are illustrated in the top row of Fig. 1. Focusing first on the middle column, where the December to February (DJF) mean is shown, we see a clear warming of the Eurasian continent. This independently confirms the result shown in Fig. 1 of Shindell et al. (2004), obtained with an earlier version of the temperature reconstruction of the CRU (Jones and Moberg, 2003), with a slightly different methodology. The post-Krakatau warming in DJF is further corroborated by the other two reconstructions we have analyzed, the NOAA and GISTEMP datasets, shown in the middle column of the bottom two rows of Fig. 1, respectively. Note, however, that the reference period for those two



datasets is shorter (only 3 winters) than the standard 5-winter period used throughout this paper, as those reconstructions do not extend far enough into the past.

One might then legitimately wonder to what degree these anomalies might depend on the length of the reference period used. To address that concern, in Fig. 2 we show the post-Krakatau anomalies computed with respect to a 3-, 5- and 10-winter reference period from the CRUTEM product, the only one for which this calculation can be performed. As one can see, the warming in DJF is very robust: averaged over the Eurasian box, we obtain a mean warming of 1.75°C, 1.89°C and 1.91°C for, respectively 3-, 5- and 10-winter reference periods. We conclude, therefore, that the post-Krakatau DJF temperature anomaly is robust in the observations, irrespective of the dataset and reference period one uses.

Having established that a relative warming over Eurasia did, in fact, occur in the first winter following the 1993 Krakatau eruption, we now ask two important questions: (1) how persistent was that anomaly, and (2) how unusual was it? To shed some light on the persistence, we have plotted in left and right columns of Fig. 1, respectively, the November to January (NDJ) and January to March (JFM) anomalies. While Eurasian warming is present in NDJ, one can see that it turns into a cooling in JFM, notably to the east of the Ural Mountains. Examination of individual months (not shown) reveal a clear cooling over the Urals starting in February, and becoming large enough in March to dominate the JFM average. This suggests that, at least for this event, the warming anomalies over Eurasia did not persist into the late winter. Note that the seasonal polar vortex breakdown typically occurs around April 15 (as estimated from reanalyses, over the period 1981-2016, by Butler et al., 2019): hence the presumed stratospheric pathway would be operative in the month of March, but it does not appear to be persistent.

Finally, focusing again on the DJF months, we address the question of whether the post-Krakatau winter warming anomalies over Eurasia were, in some way, exceptional. Recall that the 1883 eruption of Mt. Krakatau was a cataclysmic event, causing over 36,000 deaths and producing a global cooling that lasted for several years, with temperatures not returning to normal until 1888 (Judd et al., 1888; Simkin and Fiske, 1983). And two modeling studies have even claimed that Krakatau may have so strongly impacted Southern Ocean temperatures as to alter ocean heat uptake for many decades (Fyfe, 2006; Gleckler et al., 2006). In such a context, one would expect the 1883/84 Eurasian temperature anomalies to clearly stand out.

It is perhaps disappointing, therefore, to discover that the post-Krakatau anomalies are far from exceptional, when compared to all the winters available in the instrumental record, as shown in Fig. 3a. In that figure we plot the probability distribution function (PDF) of wintertime (DJF) anomalies averaged over the Eurasian box, computed using all the available winters in the CRUTEM datasets (from 1850 to present), with the post-Krakatau winter highlighted by the blue vertical line. Notice how that line falls almost evenly between the one-$\sigma$ and two-$\sigma$ (dashed) lines: with a mean value at 1.89°C, the winter 1883/84 falls within the 85th percentile of the distribution. This is confirmed by the PDFs computed from the NOAA and GISTEMP dataset, using the shorter 3-winter reference period, shown in Fig. 3b and c, respectively. In those datasets too, the anomalies fall between one and two $\sigma$ of the PDF. So, whereas the months (and years) following the 1883 Krakatau eruption may have been memorable in many respects, that adjective does not apply to the Eurasian wintertime anomalies. We emphasize that this result does *not* rely on using climate models: it is a purely observational result, and is robust across all three temperature reconstructions analyzed here.





## 4 Reanalyses

Next we turn to examining the 20CR reanalysis, for which a 56-member ensemble is available, starting from the year 1850. The top row of Fig. 4 shows the *ensemble mean* post-Krakatau Eurasian anomalies in 20CR, for NDJ, DJF and JFM, and can be directly compared with the top row in Fig. 1. Although only sea level pressure measurements are assimilated into 20CR, its temperature anomalies are in good agreement with the observations. Note, in particular, how warming in NDJ and DJF gives
way to a cooling in JFM over a large portion of Eurasia. This good agreement is perhaps not surprising, since an accurate reanalysis is expected to capture the observations.

Surprising, perhaps, is what can be seen by examining individual members of the ensemble. The anomalies for member #42 are shown in the middle row of Fig. 3: this is the member with the minimum temperature anomalies in DJF across the ensemble. Note the strong cooling in NDJ and DJF over most of Eurasia, which is diametrically opposite to the ensemble
mean. It is important to stress that member #42 is driven by the *same* natural and anthropogenic forcings (including volcanic aerosols and $CO_2$), and constrained by the *same* sea level pressure measurements, as all the other members of the ensemble. Its Eurasian cooling anomalies, therefore, are entirely consistent with observations.

For completeness, in the bottom row of Fig. 3 we show member with the maximum Eurasian temperature anomalies in DJF (member #7). The contrast with member #42 is remarkable. The key point here is that, driven with identical forcings
and observations, these two members give diametrically opposite results. The conclusion, therefore, is that the observational uncertainties are large and that, while unlikely (3 of 56 members), a post-Krakatau cooling over Eurasian is not incompatible with observations.

## 5 Models

Finally, to quantify the *volcanically forced* post-Krakatau Eurasian wintertime surface temperature anomaly (if any), i.e. to
separate it from the large internal variability, and to establish whether the stratospheric pathway mechanism may have been operative in the first winter following that eruption, we turn to climate models. We start by examining the 100-member ensemble of historical simulations made available by the Max Plank Institute for Meteorology (hereafter MPI, Maher et al., 2019). We are aware that a few, large, initial condition ensembles of historical simulations are now available (Deser et al., 2020): however, most of them do not cover the 1883 Krakatau eruption. Of the few that do, we have decided here focus on the MPI ensemble
because the underlying model is a so-called "high-top" model, with a good representation of the stratospheric circulation and, more importantly, of its variability. This is important if we one is trying to evaluate whether the stratospheric pathway mechanism is indeed affecting the post-eruption Eurasian winter surface temperatures.

We start by examining the tropical, lower-stratospheric, post-eruption, temperature anomalies in the historical MPI simulations, to ensure that the volcanic aerosols in that model do indeed produce a noticeable warming in that region after each
large, low-latitude eruption. The time series of tropical 50 hPa temperature anomalies, from 1850 to 2015, are shown in Fig. 5.





Comparison with Fig. 3 of Driscoll et al. (2012) should reassure the reader that these simulations are comparable[2] with those of the CMIP5 models. Note that the 1883/84 ensemble mean MPI anomaly in the tropical lower stratosphere is close to 8°C, with some members showing an anomaly as large[3] as 10°C. Note, in addition, that the post-Krakatau tropical warming is in this model is similar the one for the 1991 Pinatubo eruption.

With this in mind, we now examine the *forced* wintertime Eurasian temperature response to the 1883 Krakatau eruption, as simulated by the MPI ensemble, obtained by averaging all 100 members. It is shown in the top row of Fig. 6 and, as expected, it is basically zero. We emphasize that this result is consistent with the findings of PBS19 for the 1991 Pinatubo eruption, and with the above cited papers which have shown that there is *no statistically significant volcanically forced* surface winter warming in the CMIP3 and CMIP5 models. In the spirit of PBS19, and unlike most previous studies on this subject, we here

also examine individual ensemble members. In the middle and bottom rows of Fig. 6, we show the wintertime anomalies for the coldest and warmest member, respectively, of the MPI ensemble. We emphasize that these two members are subjected to an identical volcanic aerosol forcing, which causes a very large warming in the tropical lower stratosphere in both cases (see Fig. 5). And yet, member #37 simulates a large winter cooling while member #38 simulates a large winter warming over Eurasia. Confirming the results of PBS19, this clearly indicates that the large internal variability of the midlatitude circulation

completely overwhelms any potential impact from volcanic aerosols in the stratosphere for eruptions of this magnitude.

    The inability of the proposed stratospheric pathway to affect wintertime Eurasian surface temperatures in a significant way, even after a very major event such as the 1883 Krakatau eruption, is further illustrated in Fig. 7, where the zonal mean zonal wind anomalies at 60N and 10 hPa (a standard measure of the strength of the stratospheric polar vortex; see e.g. Charlton and Polvani, 2007) are plotted against the surface temperature anomalies over the Eurasian box in the DJF months following the

1883 Krakatau eruption, one dot for each of the 100 members of the ensemble. Notice first that, in the ensemble mean, the model simulates a stratospheric polar vortex deceleration of nearly 5 m/s. This independently confirms the finding of Bittner et al. (2016a): analyzing the same ensemble using a slightly different metric, they reported that "12 ensemble members are necessary to identify the mean polar vortex anomaly of approximately 4.5 m/s", when considering the Krakatau eruption alone. In other words, given a sufficiently large ensemble, a small but statistically significant acceleration of the polar vortex

*forced* by the Krakatau eruption can be established.

    However, and this is the key point of our paper: a polar vortex acceleration of a few m/s does *not* translate into a statistically significant Eurasian surface temperature anomaly. Bittner et al. (2016a) did not address that important point, which we here make explicit. As one can see in Fig. 7, the ensemble mean (i.e. the volcanically forced) surface temperature anomaly is tiny and actually negative but, most importantly, it is *not* statistically significant in this ensemble (cf. the top row of Fig. 6) and

it is uncorrelated correlated with the strength of the stratospheric polar vortex ($r \sim 0.3$, implying that 90% ot the surface temperature variance is not explained by the polar vortex strength) . This fact corroborates the finding of PBS19 for the 1991

---

[2]Unlike Driscoll et al. (2012), we do not detrend the time series in Fig. 5, as the stratospheric cooling is *not linear* over the long 1850-2005 period. Note how the cooling rate increases substantially in the second half of the 20th century, as a consequence of ozone depletion.

[3]We are well aware that most current-generation models models are believed to overestimate the amplitude of volcanic forcing of the climate system, notably in the winter seasons. For an recent appraisal of this issue, the reader may consult Chylek et al. (2020), and references therein. It should, however, be clear to the reader that this bias – if it exists – does not invalidate the main conclusion of our study: in fact, it makes it considerably *stronger*.





Pinatubo eruption: for that event too models simulate a small polar vortex acceleration in the first post-eruption winter but, again, that does not translate into a significant surface warming.

Now, one might try argue that MPI model analyzed is flawed in some way, rendering it incapable of capturing the surface
response to a small polar vortex acceleration. Our reply to that objection is threefold. First, the MPI model has been used extensively, for many years, to study stratosphere-troposphere dynamical coupling and the effect of volcanic eruptions on climate (e.g., Zanchettin et al., 2019; Illing et al., 2018; Timmreck et al., 2016; Bittner et al., 2016a, b, just to cite the most recent papers): to our knowledge, nobody has suggested a specific and demonstrable flaw that would render it inappropriate to simulate the 1883 Krakatau eruption. Second, we have separately analyzed the 1991 Pinatubo eruption in the MPI 100-member
ensemble (not shown), and found its results nearly identical to those reported in PBS19 using different models: this suggests that there is every reason to believe that the MPI results are robust. Third: the lack of a post-Krakatau surface warming response over Eurasia, is completely consistent with the many recent studies cited above showing a lack of forced Eurasia winter warming after large volcanic eruptions. Can all these models be so uniformly wrong?

Finally, to complete the picture, we analyze 140 CMIP5 historical simulations (from 40 different models), in an attempt
to determine the role (if any) of the stratospheric pathway mechanism. For purposes of understanding Eurasian wintertime post-volcanic warming, analysis of the CMIP5 models for has already been performed by several studies (e.g., Driscoll et al., 2012; Wunderlich and Mitchell, 2017) which, nearly[4] unanimously, have reported the absence of a volcanically forced winter Eurasian warming, in agreement with results from the CMIP3 (Stenchikov et al., 2006) and many other single-model studies.

Here, we take a different approach to examine the CMIP5 simulations: focusing exclusively on the 1883 Krakatau eruption,
we separately analyze the so-called "high-top" models, which have a much better representation of the stratosphere and of its variability, and the remaining "low-top" models (see Methods). One might naively expect that if the stratospheric pathway matters at all, some difference might emerge between these two group of models. As it happens, no such difference exists.

This is illustrated in Fig. 8, which compactly summarizes the findings of our study including, (1) the good agreement between observations and the 20CR reanalysis, (2) the large uncertainties in 20CR and (3) the non-existent forced response in
the MPI ensemble. The CMIP5 models are shown in the two left-most box-and-whisker plots, where the anomalies for the high-top and low-top models have been separately averaged: note that for both types, the multi-model mean post-Krakatau anomaly is *statistically insignificant*, and no obvious difference can be seen between the two types. Thus, the CMIP5 models give no[5] evidence of a stratospheric pathway playing a role in affecting Eurasian surface temperatures in the first post-Krakatau winter.


---

[4] A single dissenting voice, Zambri and Robock (2016), concluded that the CMIP5 models show evidence for a Eurasian wintertime post-volcanic warming. However, that conclusion was reached by lowering the confidence level to 90%, and the areas of significance were found to be small and did not include central Europe. Also, that result was obtained by averaging together the Pinatubo and Krakatau eruptions, so it is not directly comparable with the present study. Finally, we simply note that the findings of that study have not, to date, been independently reproduced.

[5] Charlton-Perez et al. (2013) also explored the possible existence of post-volcanic difference between the high-top and low-top models, although they examined a small subset of the CMIP5 models analyzed here (about half). Averaging together the first two winters following the 1991 Pinatubo and the 1982 El Chichón eruptions, they also found no difference in winter polar vortex anomalies between the mean of the high-top and of the low-top models.





## 6   Summary and Discussion

Building on a recent study of Eurasian winter temperature anomalies following the 1991 Pinatubo eruption (PBS19), we have here investigated those anomalies following the 1883 Krakatau eruption, examining observational reconstructions, reanalyses and model output (including one large ensemble and hundreds of CMIP5 simulations). Several key points have emerged from
our analysis (see Fig. 8):

1. Observations indicate that from November 1883 to January 1884, Eurasian surface temperatures were warmer than in the preceding winters, with a DJF mean anomaly between 1.5 and 2°C when averaged over the Eurasian continent. Such seasonal anomalies, however, fall in the 1-$\sigma$ to 2-$\sigma$ range (of the 1850-present PDF), and are therefore unexceptional.

2. A 56-member ensemble of reanalyses (20CR) reveals a very large spread in post-Krakatau Eurasian wintertime surface
temperatures anomalies, with a few members showing cooling. Given such large uncertainties, therefore, we conclude that colder than normal post-Krakatau winter conditions are not incompatible with observations.

3. Analysis of hundreds of state-of-the-art model simulations reveals the *complete absence of a volcanically forced response* over Eurasia in the first post-Krakatau winter. Furthermore, we find no evidence for a stratospheric pathway, which could (in theory) link tropical lower-stratosphere anomalies from volcanic aerosols to Eurasian surface temperature anomalies.

These new findings regarding the 1883 Krakatau eruption strongly corroborate the findings regarding the 1991 Pinatubo eruption, reported in PBS19. This is perhaps not surprising, since these two eruptions are of comparable magnitude. Nonetheless, we submit that it is important to document each eruption individually, as each one offers an independent observational data point. Each eruption is unique, in some way: the state of El Niño-Southern Oscillation (ENSO) and the quasi-biennal oscillation (QBO) indices, the specific location and month of the eruption, the vertical penetration of the volcanic gas and dust into the
stratosphere, the rate and extent of latitudinal spreading of the aerosol cloud are specific to each event, and could affect how the eruption impacts surface temperatures at higher latitudes. It is, therefore, important to have examine each eruption individually. For instance, the 1902 eruption of Santa Maria was also a powerful (VEI=6) low-latutide event, but the volcanic stratospheric sulfur injection (VSSI) from that eruption was only a fraction of the Pinatubo and Krakatau values (Toohey and Sigl, 2017), with the expectation of a comparably smaller impact at high latitudes. In contrast, the 1883 Krakatau eruption stands together
with the 1991 Pinatubo eruption as the only other recent, large (VEI=6), low-latitude, eruption with considerable stratospheric sulfur injection (for both, VSSI~ 9 Tg[S]).

We now ask, therefore: how do the findings reported above for Krakatau, taken together with the findings reported in PBS19 for Pinatubo, reshape our understanding of whether large, low-latitude eruptions can cause a winter warming over Eurasia? First, it interesting to note that an anomalous warming was indeed observed following both eruptions. However, as
we have here shown, those anomalies (~1-2°C in DJF) are far from exceptional. Second, analysis of a wealth of state-of-the-art model simulations leaves no doubt that for VEI=6 eruptions – such as Krakatau and Pinatubo – there is *no statistically significant forced* Eurasian surface winter warming in climate models. And this despite the fact that most models considerably



overestimate[6] the lower stratospheric tropical warming that follows these eruptions. Third, we emphasize that the stratospheric pathway fails to produce a significant Eurasian warming not because it is physically implausible, but because the amplitude of
the polar vortex acceleration that accompanies the warming of the lower-stratospheric by volcanic aerosols (a few meters per second, for both Pinatubo and Krakatau) is simply insufficient to overcome the very large internal variability of the midlatitude circulation.

We submit that, at this point, the evidence is overwhelming: low-latitude eruptions as large as Pinatubo or Krakatau are unable to cause a forced surface temperature anomaly over Eurasia that can be distinguished from unforced variability. As
discussed at length in PBS19, the early claims of a causal connection were based on low resolution models with poor stratospheric resolution and little stratospheric variability, and on a few observational studies which often commingled high- and low-latitude eruptions as well as the first and second post-eruption winters (we note that none of these observational studies was independently validated). The situation, therefore, is now reversed: rather than taking for granted that large, low-latitude eruptions cause Eurasian winter warming (and seeking to explain the lack of causal evidence in models as, e.g., in Driscoll
et al., 2012), the onus has now shifted to presenting evidence in support of the claim that a causal connection exists at all.

Where might one look for that evidence? If there is any hope left for the stratospheric pathway mechanism to communicate the impact of low latitude eruptions to Eurasia in winter, we need to look at larger eruptions. A few of those are well known, and the next best candidate might be the 1815 Tambora eruption. Note, however, that its VSSI index is only three times larger than Krakatau and Pinatubo (Toohey and Sigl, 2017), so even that event may not be large enough. One might then be tempted
to look further back in time, e.g. at the 1257 Samalas eruption. Doing so, however, raises additional questions. Perhaps the most troublesome is the following: dendrochronological records constitute the bulk of proxy observations used to produce temperature reconstructions, and those are mostly found in summer (the growing season). As a consequence, the robustness of winter reconstructions remains unclear (Steiger et al., 2018, for instance, show that the DJF skill scores in their reconstruction are much weaker that for the other seasons). And yet, the stratospheric pathway is fundamentally a winter mechanism, as
it requires the presence of the stratospheric polar vortex (which is disappears in the summer months). Hence, annual-mean temperatures, which are typical of many, perhaps most, reconstructions (e.g. the recent Last Millennium Climate Reanalysis Project, Hakim et al., 2016) are fundamentally inadequate for the task.

All things considered, establishing the existence of a volcanically forced winter warming over Eurasia, and of an accompanying stratospheric pathway (if any), appears to be a truly daunting challenge.

---

[6]In the case of Pinatubo, observations indicate a lower stratospheric warming of the order of 2.5°C (Labitzke and McCormick, 1992; Free and Lanzante, 2009), whereas models often show anomalies of the order of 6 to 8°C, as seen, e.g., in Fig. 5.



*Code and data availability.* All code and data are can be made available upon reasonable request.

*Author contributions.* LMP originally conceived of the study, and designed it together with SJC. SJC performed the analysis and produced the figures. LMP wrote the manuscript, in close consultation with SJC. Both authors contributed to the interpretation of the results.

*Competing interests.* The authors are not aware of competing interests.

*Acknowledgements.* LMP is grateful to the US National Science Foundation for its continued support (AGS-1914569), and wishes to thank Alan Robock and Alexandro Tejedor-Vargas for many enlightening conversations.





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



**Table 1.** Acronym, number of simulations, and high-top/low-top classification (based on Rea et al., 2018) for the historical integrations of the CMIP5 models analyzed in this paper. Information on model forcings and experimental design can be found in Taylor et al. (2012).

| model name | members | type | model name | members | type |
|---|---|---|---|---|---|
| ACCESS1.0 | 1 | LT | GISS-E2-H-CC | 1 | HT |
| ACCESS1.3 | 1 | LT | GISS-E2-R | 16 | HT |
| bcc-csm1-1 | 3 | LT | GISS-E2-R-CC | 1 | HT |
| bcc-csm1-1-m | 3 | LT | HadCM3 | 10 | LT |
| BNU-ESM | 1 | LT | HadGEM2-CC | 1 | HT |
| CCSM4 | 6 | LT | HadGEM2-ES | 4 | LT |
| CESM1-BGC | 1 | LT | inmcm4 | 1 | LT |
| CESM1-CAM5 | 3 | LT | IPSL-CM5A-LR | 5 | HT |
| CESM1-FASTCHEM | 3 | LT | IPSL-CM5A-MR | 3 | HT |
| CESM1-WACCM | 1 | HT | IPSL-CM5B-LR | 1 | HT |
| CMCC-CESM | 1 | HT | MIROC-ESM | 3 | HT |
| CMCC-CM | 1 | LT | MIROC-ESM-CHEM | 1 | HT |
| CMCC-CMS | 1 | HT | MIROC5 | 5 | LT |
| CNRM-CM5 | 10 | LT | MPI-ESM-LR | 3 | HT |
| CSIRO-Mk3-6-0 | 10 | LT | MPI-ESM-MR | 3 | HT |
| FGOALS-g2 | 4 | LT | MPI-ESM-P | 2 | HT |
| FIO-ESM | 3 | LT | MRI-CGCM3 | 5 | HT |
| GFDL-CM3 | 5 | HT | MRI-ESM1 | 1 | HT |
| GFDL-ESM2G | 3 | LT | NorESM1-M | 3 | LT |
| GISS-E2-H | 10 | HT | NorESM1-ME | 1 | LT |

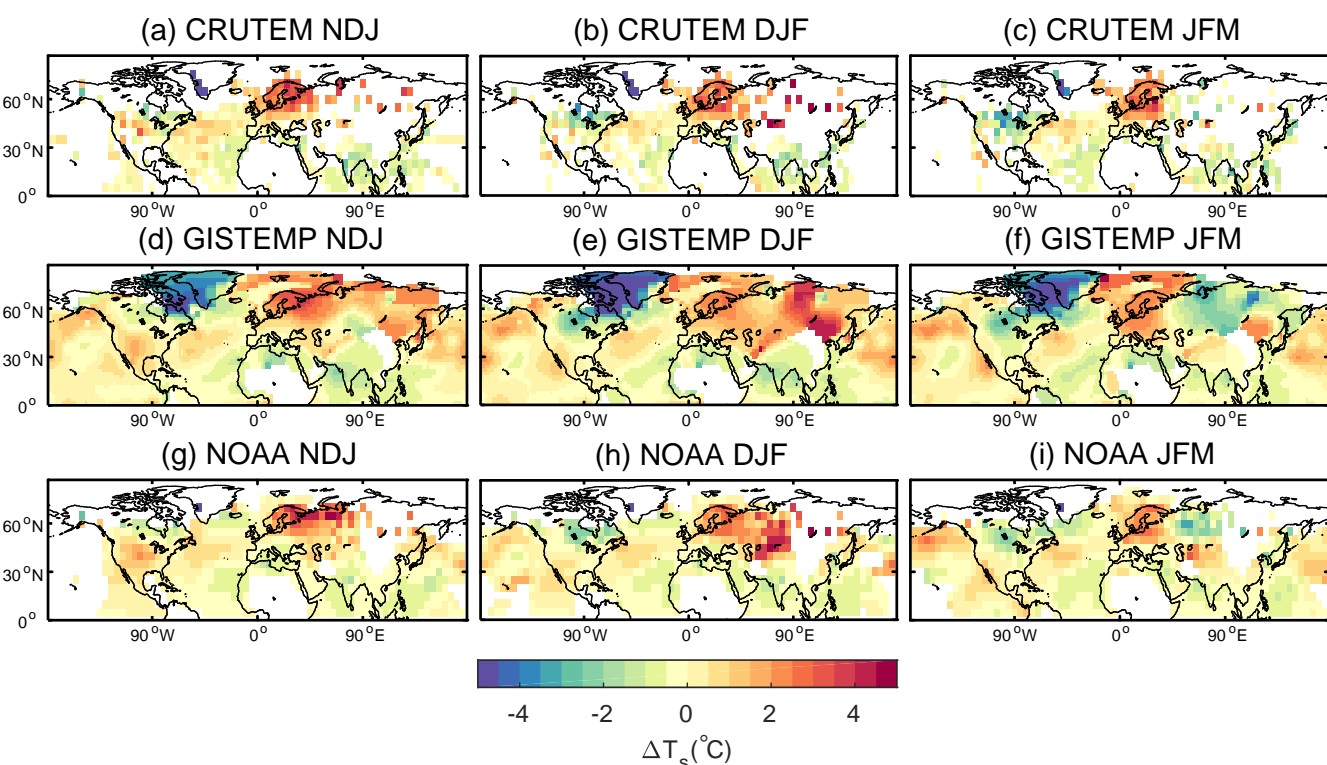

**Figure 1.** Top row: CRUTEM post-Krakatau surface temperature anomalies ($\Delta T_s$) averaged over (a) November to January (b) December to February and (c) January to March. Middle and bottom row: as top row, but for GISTEMP and NOAA. Due to data availability, the anomalies in the middle and bottom rows are computed using a shorter 3-winter reference period.



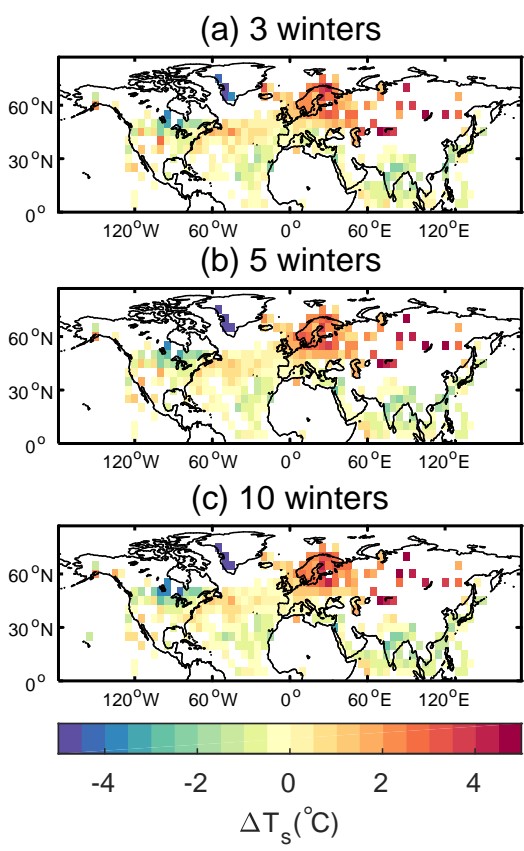

**Figure 2.** Post-Krakatau CRUTEM surface temperature anomalies ($\Delta T_s$) in DJF, using (a) three (b) five and (c) ten winters for the reference period.



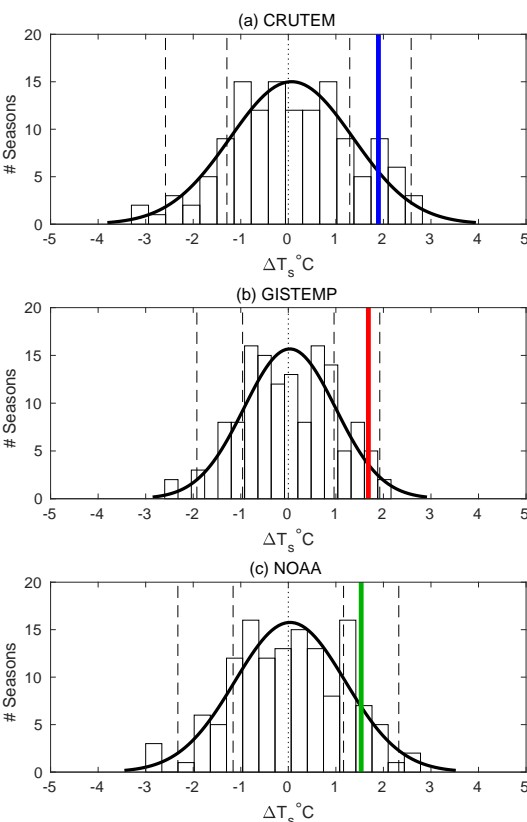

**Figure 3.** (a) Black boxes: histogram of CRUTEM post-Krakatau Eurasian temperature anomalies ($\Delta T_s$) in DJF, over the period 1850 to present. Colored bar: the 1883/84 post-Krakatau anomalies. Solid black line: Gaussian fit; dashed black lines: the 1-$\sigma$ and 2-$\sigma$ intervals. (b) and (c): as in (a) but for GISTEMP and NOAA, respectively, using the shorter 3-winter reference period.

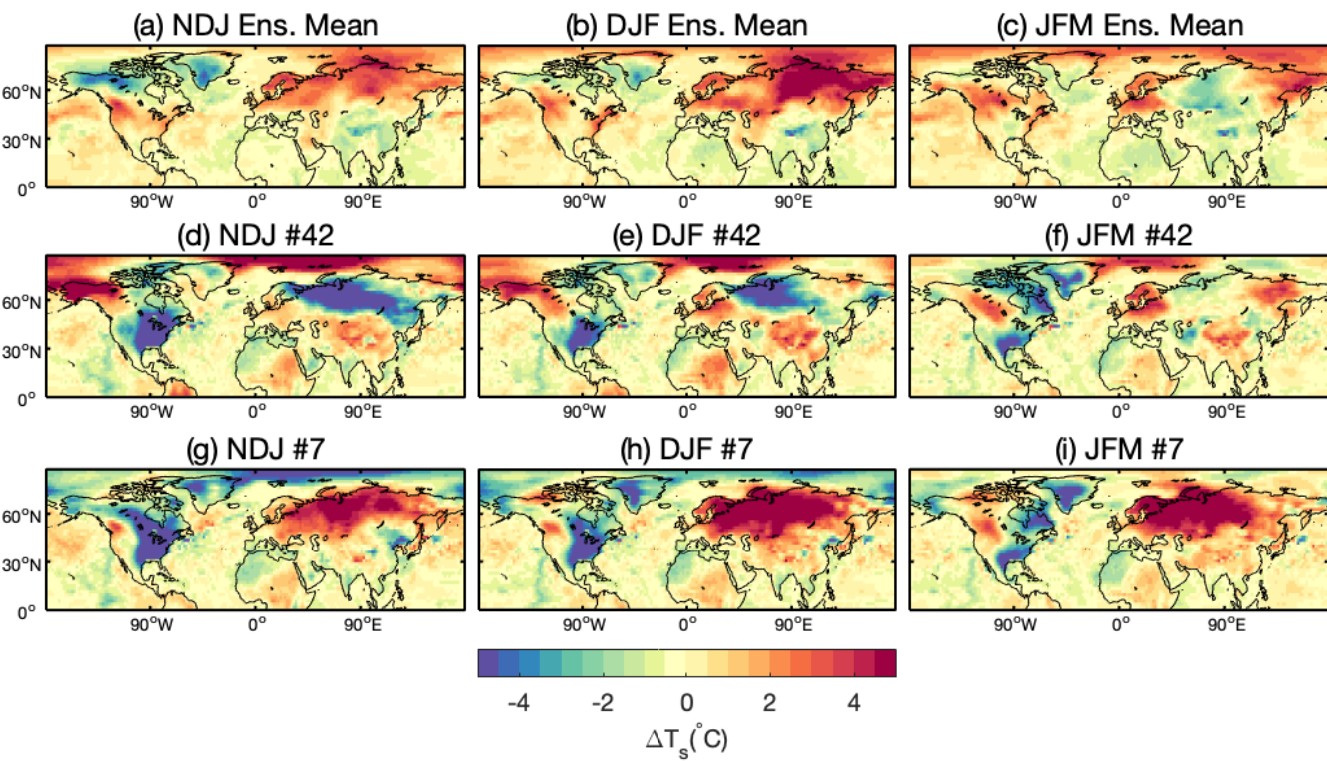

**Figure 4.** Top row: 20CR ensemble mean, post-Krakatau, surface temperature anomalies ($\Delta T_s$) averaged over (a) November to January (b) December to February and (c) January to March. Middle row: as in the top row, but for ensemble member #42. Bottom row: as in the top row, but for ensemble member #7. Members 42 and 7 have, respectively, the minimum and maximum post-Krakatau Eurasian DJF anomalies across the 56-member 20CR ensemble.

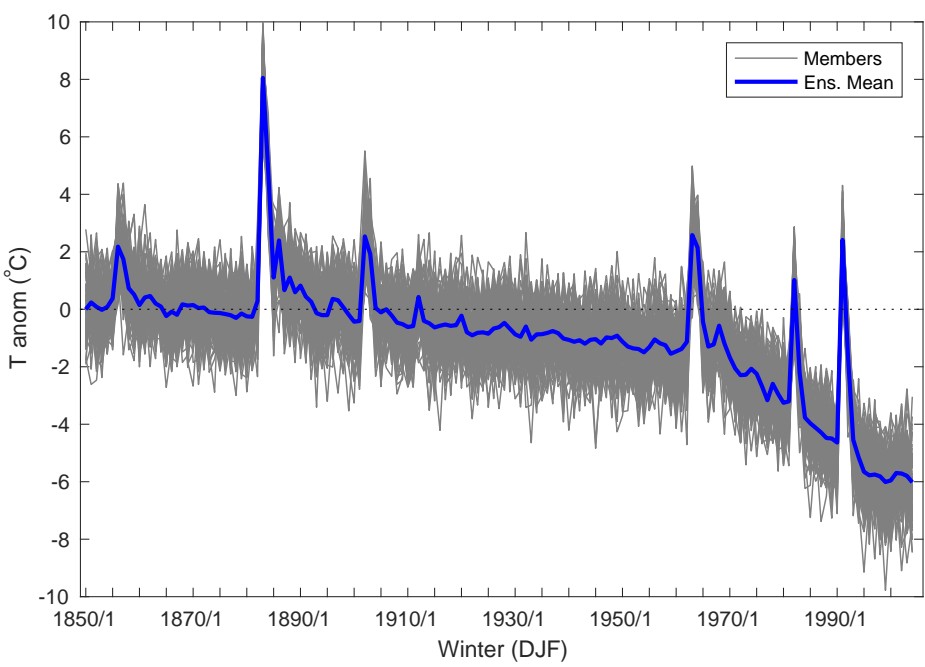

**Figure 5.** Tropical (30°S-30°N) temperature anomalies in DJF at 50hPa, in the MPI historical runs simulations, from 1850/51 to 2004/5. Grey line: individual members; blue line: ensemble mean. Anomalies are here defined based on the 1861-1880 period.



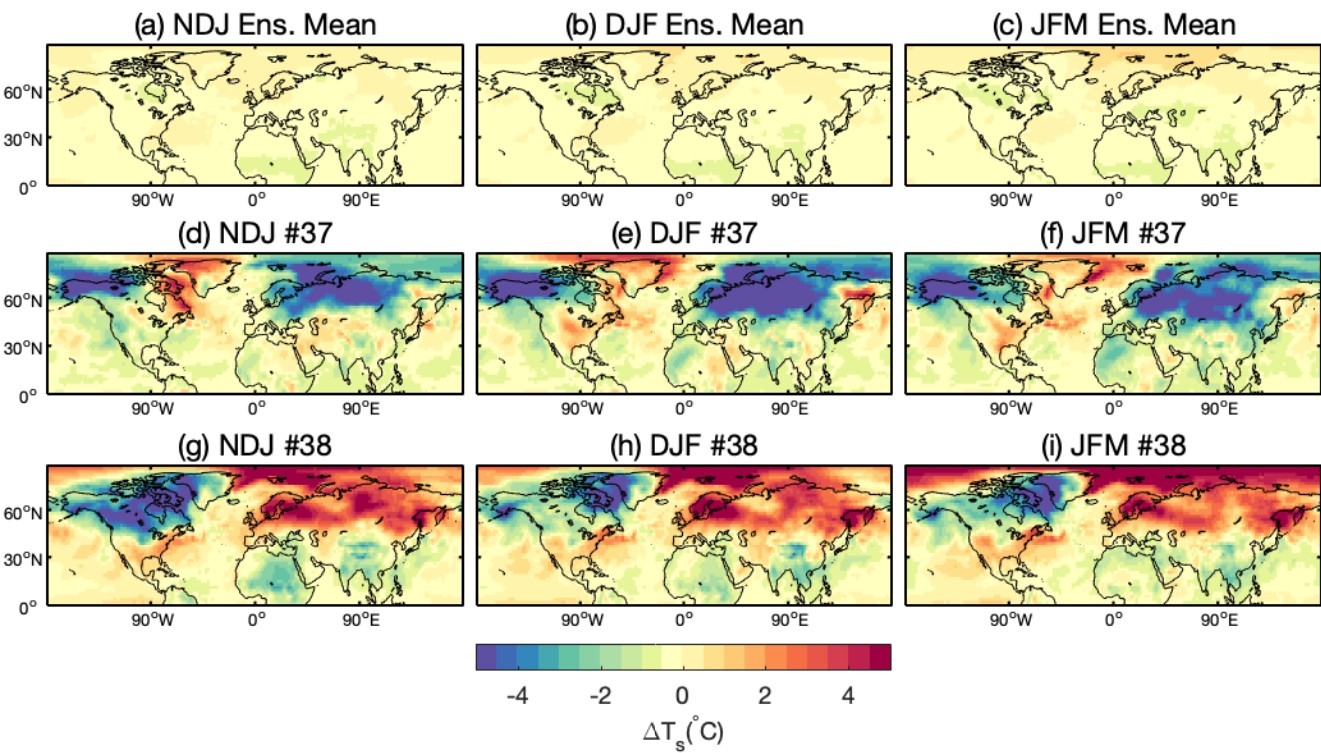

**Figure 6.** Top row: MPI ensemble mean, post-Krakatau, surface temperature anomalies ($\Delta T_s$) averaged over (a) November to January (b) December to February and (c) January to March. Middle row: as in the top row, but for ensemble member #37. Bottom row: as in the top row, but for ensemble member #38. Members 37 and 28 simulated, respectively, the minimum and maximum post-Krakatau Eurasian DJF anomalies across the 100-member MPI ensemble.

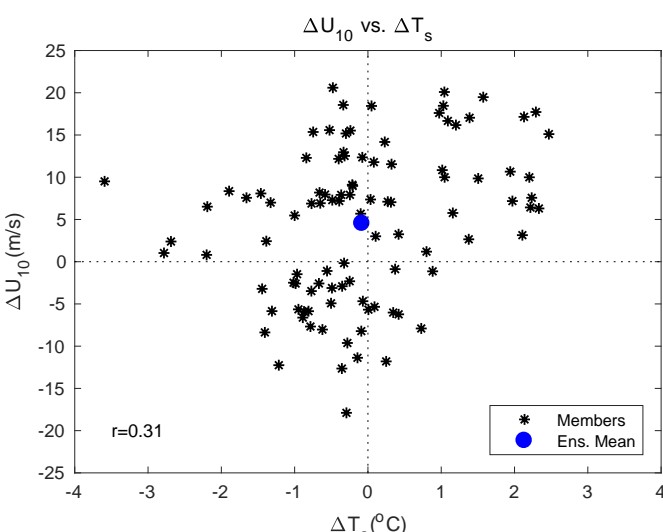

**Figure 7.** Post-Krakatau, DJF, zonal mean, zonal wind anomalies at 10 hPa and $60°N$ ($\Delta U_{10}$) in the 100-member MPI ensemble vs the Eurasian surface temperature anomalies ($\Delta T_s$). Blue dot: ensemble mean; black asterisks: individual members.

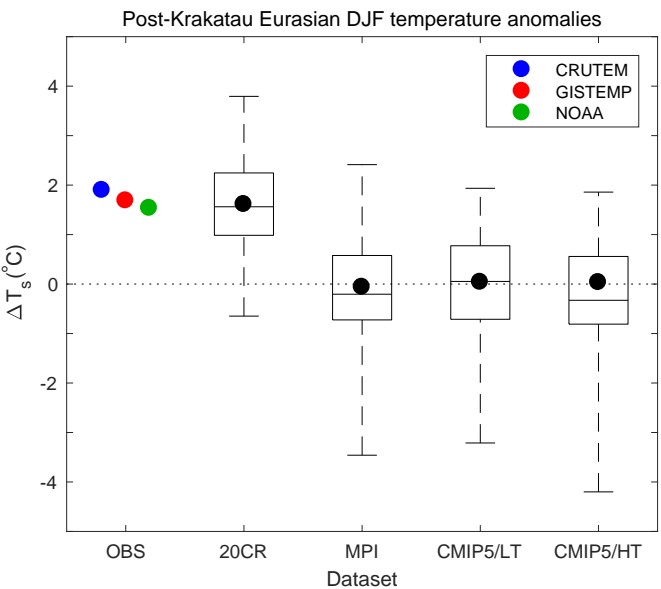

**Figure 8.** Eurasian surface temperature anomalies in the first post-Krakatau DJF season. Colored dots: observed values from CRUTEM, GISTEMP and NOAA anomalies (for the latter 2, a shorter only 3 reference period was used, owing data availability). The box plots show the median, the 25th and 75th percentiles in each dataset, while the whiskers show the spread of the ensembles for each dataset. For 20CR, a 56-member ensemble was analyzed; for the MPI model, a 100-member ensemble; for the CMIP5 low-top (LT) models 68 simulations from 21 LT models, and for the high-top (HT) models, 62 simulations from 19 HT models. See Table,1 for the list of CMIP5 HT and LT models, and the respective simulations.