# Peer review of "Scant evidence for a volcanically forced winter warming over Eurasia following the Krakatau eruption of August 1883"

_Atmospheric Chemistry and Physics, 2020_

## Referee Comment (RC1) · Anonymous Referee #1 · 17 Jul 2020

The Authors present a case study of the the Krakatau eruption of August 1883 and its impact on wintertime temperatures in the Eurasian region. Compelling evidence from observations and simulations is provided to support the conclusion that the warming in the area following Krakatau was unrelated to the eruption.

The paper is very interesting, well-written and to-the-point, and a pleasure to read. It is well suited for ACP and provides enough new scientific information to justify publication. Overall, the paper presents a nice piece of work.

I have one specific comment. In the Summary/Discussion, the Authors generally note that each eruption is unique, and list some affecting factors. However, the impact of

these in the presented case is not discussed. This should be done because the Authors are making rather general conclusions based on a single event. I would be particularly interested in some elaboration on the effect of QBO and ENSO, because there are studies that seem to point out their importance for the "stratospheric pathway" and NAO modulation, e.g. for the top-down solar influence. At least, the Authors should state the conditions during the Krakatau eruption and make a comment on the possible implications regarding their conclusions. Also, were these conditions similar during the 1992 Pinatubo eruption? I am looking forward to the Authors' response on this.

Minor corrections:

a) Page 3, line 77: check the years, 1978 should be 1878.

b) Page 8, line 240: "uncorrelated correlated" should be "weakly correlated".

---

## Referee Comment (RC2) · Matthew Toohey (Referee) · 31 Jul 2020

General comments:

This paper focuses on the "volcanic winter warming" theory that stratospheric aerosols from volcanic eruptions cause changes in atmospheric circulation, that lead in turn to warming over the northern Eurasian continent in the 1 or 2 winters after the eruption. The study looks specifically at the 1883 eruption of Krakatau, and presents analysis of surface temperature reconstructions, reanalyses and climate model output. The conclusions of the study are very similar to those of an earlier study (Polvani et al., 2019) which focused exclusively on the 1991 eruption of Pinatubo. The main conclusion of

the paper is that "the observed warming over Eurasia in the winter of 1883/84 was, in all likelihood, unrelated to the Krakatau eruption". Taken together with the prior paper on Pinatubo, the authors argue that volcanic winter warming is not a real phenomenon for eruptions of this magnitude, and that the warm Eurasian temperatures in the winters after these two eruptions were chance occurrences resulting from natural climate variability.

The paper is provocatively written, and indeed a major promise of the study is its direct challenge of the winter warming theory. Where earlier studies have raised doubts about selected components of the overall theory, this study aims to call into question its very validity. The topic is certainly open for scientific debate, and there is room for critical perspective.

However, this study contains numerous fallacies which undermine the logical argumentation. Most generally there are two main problems. First, the authors disregard the observational basis of the winter warming theory–it is mentioned in passing only once in the introduction. The fact that models do not reproduce the expected winter warming signal is perplexing, even disappointing, but it is no reason on its own to disbelieve observations. Secondly, the potential influence of volcanic aerosol on circulation or continental surface temperatures–in the single realization of reality–cannot be assessed by focusing on a single eruption. The observational basis for the winter warming theory has established that the signal is within the range of natural variability but is detectable because of its consistency across eruptions: the observational studies identified winter warming or positive NAO anomalies by compositing observations after more than 10 eruptions (e.g., Robock and Mao, 1992; Christiansen, 2008). It is only because of the statistical significance of the observed winter warming across many eruptions that one may interpret the modest observed Eurasian warming after Krakatau as being linked to the eruption. By focusing on a single eruption, and by neglecting the observational basis, the study fails to mount a valid challenge to the volcanic winter warming theory.

Specific comments:

L6: The first main finding is that "observed post-Krakatau winter warming over Eurasia was unremarkable (only between 1- and 2-sigma of the distribution from 1850 to present)." However, prior studies do not suggest that warming over Eurasia after any single eruption is necessarily remarkable. Robock and Mao (1992) show temperature anomalies from 12 eruptions since 1883: Eurasian anomalies rarely exceed +3C, are in some cases negligible, and on average suggest a mean warming of $\sim$1-2C. As another example, Christiansen (2008) showed that in the winters after 13 eruptions from 1880-2000, 11 winters showed a positive NAO. The NAO magnitude in each of those years is unremarkable–and the mean NAO anomaly is a very pedestrian $\sim$0.6–but what is remarkable is the consistency of the post-eruption anomaly. The repeated description of the Krakatau winter warming as "unremarkable" is in no way evidence against the winter warming theory, in fact, that amount of warming is quite consistent with what one would expect based on observational studies.

L7: The second finding is that "reanalyses indicate the existence of very large uncertainties, so much so that a Eurasian cooling is not incompatible with observations". This finding does not follow from the results shown. First, the phrase "not incompatible with observations" obscures the fact that only 3 out of 56 ensemble members in the reanalysis produce negative Eurasian temperature anomalies. Based on the ensemble, one should conclude that Eurasian cooling was very unlikely. Secondly, the statement refers vaguely to "observations" that "a European cooling is not incompatible with", without specifying that these observations are only the surface pressure observations that are assimilated into the reanalysis. Without more careful language, a reader might easily understand that a European cooling is not incompatible with all observations. But given the surface temperature reconstructions based on temperature measurements described in the study, this is clearly wrong. Overall, the conclusion seems to be an attempt to decrease confidence in the observation of a Eurasian winter warming after Krakatau, but there just isn't any reasonable way that results from 3/56 ensemble members from a reanalysis assimilating surface pressure measurements can have any influence on our understanding of Eurasian temperatures which are based primarily on

actual temperature measurements. In contrast, the fact that the reanalysis ensemble mean "temperature anomalies are in good agreement with the observations" can only increase our confidence that Eurasian temperatures in winter 1883/84 were indeed warmer than normal.

L8: The crux of the author's argument then comes down to the third finding, which is that "models robustly show the complete absence of a volcanically forced Eurasian winter warming". Based on this finding, the authors later conclude that "low-latitude eruptions as large as Pinatubo or Krakatau are unable to cause a forced surface temperature anomaly over Eurasia that can be distinguished from unforced variability". While the absence of winter warming in present-day model simulations is perplexing, model results cannot be used as a basis to discount a theory based on observations. The authors barely mention the observational basis of the theory, focusing more on describing the "stratospheric pathway" mechanism proposed by the early studies. This represents a "straw man" fallacy: the author's attack on the "stratospheric pathway" mechanism is justified, but this is not a valid argument against the observation-based winter warming theory itself.

L17: No justification needs to be given for a summary of background literature. It may be true that extant literature is "confusing and often contradictory", but this is hardly unique to this scientific topic, and to label prior work so might come across to some readers as a rhetorical tactic to undermine confidence in prior work.

L24: This "in a nutshell" description of the stratospheric pathway cites only papers from the 1990s, and neglects recent work that has both challenged the simple "meridional temperature gradient" mechanism and investigated other mechanisms, including planetary scale waves and tropospheric eddies (e.g., Toohey et al., 2014, Bittner et al., 2016, DallaSanta et al., 2019).

L32: It is not true that "very little aerosol is left in the stratosphere in the second post-eruption winter". Satellite-based retrievals of stratospheric aerosol optical depth (AOD)

after Pinatubo show that the peak in global mean AOD was around winter 1991/92, with a value of ∼0.1. One year later, the AOD is around 0.68, still elevated by an order of magnitude above the background value of ∼0.06. A similar result can be found if one looks particularly at the AOD in the tropical region. There may be valid pragmatic reasons to limit focus to the first post-eruption winter, but the statement that there is "very little aerosol is left in the stratosphere in the second post-eruption winter" is not true, and this claim should not be used to invalidate prior studies which averaged 2 post eruption winters.

Ll34: Stating that these papers all failed to replicate the results of earlier studies is very much oversimplifying their work. For example, Bittner et al. write: "For eruptions of the size of Krakatau and Pinatubo, the multi-model ensemble shows a strengthening of the polar vortex in the first post-eruption mid-winter, which challenges the assumption of a general failure of coupled climate models to simulate the dynamical response to volcanic eruptions." Also, Wunderlich and Mitchell (2017) do not present any results from CMIP5 models regarding the winter warming: they explore winter warming in reanalyses and present from some CMIP5 models simulated tropical temperature anomalies.

L41: While some of the earliest model studies used very small ensemble sizes, most past studies used ensembles of some reasonable number and quantified the statistical significance of the ensemble mean response, taking into account natural variability. It is therefore unjustified to claim that "much of the earlier literature had failed to properly account for the large internal variability associated with the stratospheric polar vortex and with the NAO".

L164: The number of deaths caused the Krakatau eruption has absolutely no bearing on the expected relative magnitude of the winter warming signal, as the number of deaths depends strongly on the population living in proximity to the volcano.

L224: The response of the two most extreme ensemble members illustrates that there is natural variability, and that the anomaly in any one post-volcanic year may vary from

case-to-case, but it does not negate the possibility of a non-zero mean response, i.e., a higher probability of either negative or positive anomaly.

L231: "acceleration"

L257: Wunderlich and Mitchell didn't look at winter warming in the CMIP5 models.

L258: This misrepresents the results of Stenchikov et al. (2006) who state in their abstract "The IPCC models tend to simulate a positive phase of the Arctic Oscillation in response to volcanic forcing similar to that typically observed. However, the associated dynamic perturbations and winter surface warming over Northern Europe and Asia in the post-volcano winters is much weaker in the models than in observations."

Footnote 4: For the sake of balanced consideration of prior work, reference to Zambri and Robock (2016) should come in the introduction rather than here near the end of the paper. Also, editorial commentary characterizing the work as "a single dissenting voice" or "not, to date, ... independently reproduced" is clearly rhetoric meant to undermine confidence in this study, and would benefit from being recast in more objective and quantitative terms.

References:

Bittner, M., Timmreck, C., Schmidt, H., Toohey, M. and Krüger, K.: The impact of wave-mean flow interaction on the Northern Hemisphere polar vortex after tropical volcanic eruptions, J. Geophys. Res., 121(10), doi:10.1002/2015JD024603, 2016.

Christiansen, B.: Volcanic Eruptions, Large-Scale Modes in the Northern Hemisphere, and the El Niño–Southern Oscillation, J. Clim., 21(5), 910, doi:10.1175/2007JCLI1657.1, 2008.

DallaSanta, K., Gerber, E. P. and Toohey, M.: The Circulation Response to Volcanic Eruptions: The Key Roles of Stratospheric Warming and Eddy Interactions, J. Clim., JCLI-D-18-0099.1, doi:10.1175/JCLI-D-18-0099.1, 2018.

Robock, A. and Mao, J.: Winter warming from large volcanic eruptions, Geophys. Res. Lett., 19(24), 2405, doi:10.1029/92GL02627, 1992.

Stenchikov, G., Hamilton, K., Stouffer, R. J., Robock, A., Ramaswamy, V., Santer, B. and Graf, H.-F.: Arctic Oscillation response to volcanic eruptions in the IPCC AR4 climate models, J. Geophys. Res., 111(D7), doi:10.1029/2005JD006286, 2006.

Toohey, M., Krüger, K., Bittner, M., Timmreck, C. and Schmidt, H.: The impact of volcanic aerosol on the Northern Hemisphere stratospheric polar vortex: mechanisms and sensitivity to forcing structure, Atmos. Chem. Phys., 14(23), 13063–13079, doi:10.5194/acp-14-13063-2014, 2014.

Wunderlich, F. and Mitchell, D. M.: Revisiting the observed surface climate response to large volcanic eruptions, Atmos. Chem. Phys., 17(1), 485–499, doi:10.5194/acp-17-485-2017, 2017.

Zambri, B. and Robock, A.: Winter warming and summer monsoon reduction after volcanic eruptions in Coupled Model Intercomparison Project 5 (CMIP5) simulations, Geophys. Res. Lett., 43(20), 10,920-10,928, doi:10.1002/2016GL070460, 2016.

---

## Author Comment (AC1)

(The authors' replies to the referees are in bold)
* * *
REPLY TO REFEREE #1
* * *
The Authors present a case study of the the Krakatau eruption of August 1883 and its
impact on wintertime temperatures in the Eurasian region. Compelling evidence from
observations and simulations is provided to support the conclusion that the warming in
the area following Krakatau was unrelated to the eruption.

The paper is very interesting, well-written and to-the-point, and a pleasure to read. It is
well suited for ACP and provides enough new scientific information to justify publication.
Overall, the paper presents a nice piece of work.

I have one specific comment. In the Summary/Discussion, the Authors generally note
that each eruption is unique, and list some affecting factors. However, the impact of
these in the presented case is not discussed. This should be done because the Authors
are making rather general conclusions based on a single event. I would be particularly
interested in some elaboration on the effect of QBO and ENSO, because there are
studies that seem to point out their importance for the "stratospheric pathway" and
NAO modulation, e.g. for the top-down solar influence. At least, the Authors should
state the conditions during the Krakatau eruption and make a comment on the possible
implications regarding their conclusions. Also, were these conditions similar during the
1992 Pinatubo eruption? I am looking forward to the Authors' response on this.

**We thank the referee for the kind words.  Following his/her suggestion, we have added a
new paragraph to the last section, discussing the potential QBO/ENSO effects on both the
Krakatau and Pinatubo eruptions.**

Minor corrections:
a) Page 3, line 77: check the years, 1978 should be 1878.

**Yes, this was a typo.  We have fixed it. Thank you.**

b) Page 8, line 240: "uncorrelated correlated" should be "weakly correlated".

**Corrected.  Thank you.**
* * *
REPLY TO REFEREE #2
* * *
This paper focuses on the "volcanic winter warming" theory that stratospheric aerosols
from volcanic eruptions cause changes in atmospheric circulation, that lead in turn to
warming over the northern Eurasian continent in the 1 or 2 winters after the eruption.
The study looks specifically at the 1883 eruption of Krakatau, and presents analysis of
surface temperature reconstructions, reanalyses and climate model output. The con-
clusions of the study are very similar to those of an earlier study (Polvani et al., 2019)
which focused exclusively on the 1991 eruption of Pinatubo. The main conclusion of
the paper is that "the observed warming over Eurasia in the winter of 1883/84 was, in
all likelihood, unrelated to the Krakatau eruption". Taken together with the prior paper
on Pinatubo, the authors argue that volcanic winter warming is not a real phenomenon
for eruptions of this magnitude, and that the warm Eurasian temperatures in the win-
ters after these two eruptions were chance occurrences resulting from natural climate
variability.

The paper is provocatively written, and indeed a major promise of the study is its direct
challenge of the winter warming theory. Where earlier studies have raised doubts about
selected components of the overall theory, this study aims to call into question its very
validity. The topic is certainly open for scientific debate, and there is room for critical
perspective.

However, this study contains numerous fallacies which undermine the logical argumen-
tation. Most generally there are two main problems. First, the authors disregard the
observational basis of the winter warming theory--it is mentioned in passing only once
in the introduction. The fact that models do not reproduce the expected winter warming
signal is perplexing, even disappointing, but it is no reason on its own to disbelieve ob-
servations. Secondly, the potential influence of volcanic aerosol on circulation or conti-
nental surface temperatures--in the single realization of reality--cannot be assessed by
focusing on a single eruption. The observational basis for the winter warming theory
has established that the signal is within the range of natural variability but is detectable
because of its consistency across eruptions: the observational studies identified winter
warming or positive NAO anomalies by compositing observations after more than 10
eruptions (e.g., Robock and Mao, 1992; Christiansen, 2008). It is only because of the
statistical significance of the observed winter warming across many eruptions that one
may interpret the modest observed Eurasian warming after Krakatau as being linked

to the eruption. By focusing on a single eruption, and by neglecting the observational basis, the study fails to mount a valid challenge to the volcanic winter warming theory.

**We are grateful to Dr. Toohey for a careful reading of our manuscript.  We are delighted to read that he sees nothing wrong with the methodology, or with the results of our study: he simply questions our interpretation.  Specifically, he notes "two main problems", which we here address in turn.**

**(1) We do not, in the least, disregard the "observational evidence": we consider it unreliable.  We refer to it twice in the paper: in the introduction section, and again in the conclusion section.  In our previous paper (Polvani et al 2019, hereafter PBS19) we have discussed in great detail why the claims made by the handful of papers which have proposed the winter warming theory are, in our opinion, not robust .  We saw no reason to repeat that material here.  However, for the sake of completeness, in the revised manuscript we have now explicitly pointed the reader to the discussion in PBS19.  It can be found on line 319 of the revised manuscript.**

**(2) Secondly, the referee claims that "the potential influence of volcanic aerosol... cannot be assessed by focusing on a single eruption."  We could not agree more.  Having analyzed the 1992 Pinatubo eruption in our earlier paper (PBS19), in this manuscript we study in detail the next big eruption for which we have good observations: the 1883 Krakatau eruption.  In fact, in upcoming papers, we will report on earlier eruptions as well.  However, we firmly believe that averaging over many eruptions -- especially over eruptions of different MAGNITUDES (as done in nearly all papers in this subject) -- is fundamentally incorrect.  In seeking a forced response, the AMPLITUDE of the forcing matters.  One would never think of averaging the responses in an RCP4.5 scenario with those of an RCP8.5 scenario.  So, why do some many papers in this field blithely average responses from volcanoes of greatly different magnitudes (e.g. Tambora and El Chichon, which differ by more than a factor of 4)?**

**For the moment we have carefully analyzed -- separately -- Pinatubo (in PBS19) and Krakatau (in this paper), and the results are very consistent.  Therefore, in the final section of the paper, we draw the appropriate conclusions from these.  We do not believe there are any "fallacies" in our methodology or our reasoning.**

Specific comments:

L6: The first main finding is that "observed post-Krakatau winter warming over Eurasia was unremarkable (only between 1- and 2-sigma of the distribution from 1850 to present)." However, prior studies do not suggest that warming over Eurasia after any single eruption is necessarily remarkable. Robock and Mao (1992) show temperature anomalies from 12 eruptions since 1883: Eurasian anomalies rarely exceed +3C, are in some cases negligible, and on average suggest a mean warming of ~1-2C. As another example, Christiansen (2008) showed that in the winters after 13 eruptions from 1880-2000, 11 winters showed a positive NAO. The NAO magnitude in each of those years is unremarkable--and the mean NAO anomaly is a very pedestrian ~0.6--but what is remarkable is the consistency of the post-eruption anomaly. The repeated description of the Krakatau winter warming as "unremarkable" is in no way evidence against the winter warming theory, in fact, that amount of warming is quite consistent with what one would expect based on observational studies.

**The reason we employ the word "unremarkable" is that previous studies to date have not placed the post-eruption warming in the context of natural variability, while at the same time making big claims about the importance of volcanic eruptions in affecting winter surface temperatures at high latitudes.  Need we remind the referee that Alan Robock had a paper in Science (2002) touting the Pinatubo eruption as a prime example of the warming theory, when observations clearly show the stratospheric polar vortex was not even anomalously strong that winter (as noted in later studies)?**

**Now, the referee argues that "what is remarkable is the consistency of the post-eruption anomalies".  But is that really so?  As already noted, we do not believe those claims are robust.  Let us just deal with Robock and Mao (1992), the first to propose the warming theory.  Did the referee notice that 6 of the 12 eruptions in that paper are not even in the tropics?  Why did those authors mix in so many high-latitude eruptions for which the stratospheric pathway does not apply?  And why did they pick the first winter for some eruptions and the second for some other eruptions?  One could go on...**

**In any case: the point we wish to stress in our paper is that, when a post-eruption surface warming is seen (as in the case of Krakatau) the amplitude of that warming is NOT LARGE compared to the internal variability.  This is what we are adding to the discussion in the first section of our paper: we show here that the post-Krakatau anomalies are largely INDISTINGUISHABLE from natural variability, and thus they are unremarkable.  In simpler terms: for those living in Eurasia, the post-Krakatau winter would look little different from many other warmer-that-average winters which were not preceded an eruption.**

**In any case, the referee agrees with us that the anomalies are unremarkable, and requests no correction.  So we have made no changes to the manuscript.**

L7: The second finding is that "reanalyses indicate the existence of very large uncertainties, so much so that a Eurasian cooling is not incompatible with observations". This finding does not follow from the results shown. First, the phrase "not incompatible with observations" obscures the fact that only 3 out of 56 ensemble members in the reanalysis produce negative Eurasian temperature anomalies. Based on the ensemble, one should conclude that Eurasian cooling was very unlikely. Secondly, the statement refers vaguely to "observations" that "a European cooling is not incompatible with", without specifying that these observations are only the surface pressure observations that are assimilated into the reanalysis. Without more careful language, a reader might easily understand that a European cooling is not incompatible with all observations. But given the surface temperature reconstructions based on temperature measurements described in the study, this is clearly wrong. Overall, the conclusion seems to be an attempt to decrease confidence in the observation of a Eurasian winter warming after Krakatau, but there just isn't any reasonable way that results from 3/56 ensemble members from a reanalysis assimilating surface pressure measurements can have any influence on our understanding of Eurasian temperatures which are based primarily on actual temperature measurements. In contrast, the fact that the reanalysis ensemble mean "temperature anomalies are in good agreement with the observations" can only increase our confidence that Eurasian temperatures in winter 1883/84 were indeed warmer than normal.

**We appreciate the correction: we have rephrased this sentence, and similar sentences occurring later in the paper, to make it more precise.  The point here is not "to decrease confidence in the observation of a Eurasian winter warming after Krakatau", but to "to decrease confidence in the fact the warming was caused by circulation changes, and thus by the NAO, and thus by the polar vortex, and thus by the volcanic aerosols". This is the point we are trying to demonstrate, and the 20CR reanalyses definitely add evidence to support our claim.  We apologize for not stating this accurately.**

L8: The crux of the author's argument then comes down to the third finding, which is that "models robustly show the complete absence of a volcanically forced Eurasian winter warming". Based on this finding, the authors later conclude that "low-latitude eruptions as large as Pinatubo or Krakatau are unable to cause a forced surface temperature anomaly over Eurasia that can be distinguished from unforced variability". While the absence of winter warming in present-day model simulations is perplexing, model results cannot be used as a basis to discount a theory based on observations. The authors barely mention the observational basis of the theory, focusing more on describing the "stratospheric pathway" mechanism proposed by the early studies. This represents a "straw man" fallacy: the author's attack on the "stratospheric pathway" mechanism is justified, but this is not a valid argument against the observation-based winter warming theory itself.

**First, we disagree with the statement that "the author's argument then comes down to the third finding". The first and second findings add much evidence which has not, to date, been presented.  Second, the modeling evidence for a lack of surface warming in state-of-the-art models is OVERWHELMING, and it is NOT PERPLEXING at all.  It makes a clear case for the fact that the stratospheric pathway is not operative in the models. The referee again brings up "the observational basis of the theory" which, as we have already discussed is highly questionable.  But neither models nor observations indicate the presence of that pathway. So there is no "straw man" here: there is simply an emperor with no clothes!**

L17: No justification needs to be given for a summary of background literature. It may be true that extant literature is "confusing and often contradictory", but this is hardly unique to this scientific topic, and to label prior work so might come across to some readers as a rhetorical tactic to undermine confidence in prior work.

**The referee is correct: there are many scientific topics with a literature full of claims and counterclaims.  But that very fact speaks for itself.  All papers agree that increasing CO2 warms the earth, and that makes it a well established fact.  Conversely, the huge "confusion and contradiction" that plagues the volcanic warming theory literature clearly indicates that it is NOT well established at all.  We think it is important to emphasize this, to encourage the colleagues to critically read the published literature.**

L24: This "in a nutshell" description of the stratospheric pathway cites only papers from the 1990s, and neglects recent work that has both challenged the simple "meridional temperature gradient" mechanism and investigated other mechanisms, including planetary scale waves and tropospheric eddies (e.g., Toohey et al., 2014, Bittner et al., 2016, DallaSanta et al., 2019).

**We have cited the seminal papers that propounded the "stratospheric pathway theory". Several subsequent papers, realizing that the original theory did not work, invoked progressively more complicated and unlikely mechanisms, none of which has proven robust. For instance, the tropospheric "meridional temperature gradient" theory proposed by Stenchikov et al (2002) has been invalidated by DallaSanta et al (2019), who found that "a naive argument that the stratospheric warming increases the equator-to-pole temperature gradient (and so strengthens the polar vortex) cannot qualitatively predict the simulated response".  In the introduction to our paper, we do not think it is helpful to confuse the reader with these later claims and counterclaims.**

L32: It is not true that "very little aerosol is left in the stratosphere in the second post-eruption winter". Satellite-based retrievals of stratospheric aerosol optical depth (AOD) after Pinatubo show that the peak in global mean AOD was around winter 1991/92, with a value of ~0.1. One year later, the AOD is around 0.68, still elevated by an order of magnitude above the background value of ~0.06. A similar result can be found if one looks particularly at the AOD in the tropical region. There may be valid pragmatic reasons to limit focus to the first post-eruption winter, but the statement that there is "very little aerosol is left in the stratosphere in the second post-eruption winter" is not true, and this claim should not be used to invalidate prior studies which averaged 2 post eruption winters.

**"Very small" may not be the correct expression, so the referee has a point. However, the fact remains that the amount of volcanic aerosols left in the second winter is a small fraction of the one present in the first winter. We have rephrased this sentence to make that clear. We are grateful for the correction.**

L34: Stating that these papers all failed to replicate the results of earlier studies is very much oversimplifying their work. For example, Bittner et al. write: "For eruptions of the size of Krakatau and Pinatubo, the multi-model ensemble shows a strengthening of the polar vortex in the first post-eruption mid-winter, which challenges the assumption of a general failure of coupled climate models to simulate the dynamical response to volcanic eruptions." Also, Wunderlich and Mitchell (2017) do not present any results from CMIP5 models regarding the winter warming: they explore winter warming in reanalyses and present from some CMIP5 models simulated tropical temperature anomalies.

**We beg to differ. The early papers reported A CLEAR AND ROBUST WINTER WARMING at the surface following volcanic eruption. Bittner et al (2016), in contrast, do not show or even discuss the surface warming in their model (their entire paper is narrowly confined to the stratosphere). Why? If one looks at the Ph.D. thesis of Dr. Bittner, one discovers the fact -- surely deeply embarrassing to the authors, and therefore hidden away on Figure 6.4 on page 85 of that dissertation -- that the their model shows NO FORCED SURFACE WARMING IN WINTER over Eurasia after the Pinatubo eruption, even after averaging 100 simulations. That work, just to mention one example cited by the referee, TOTALLY FAILED to replicate the early observational studies and modeling studies. So, with all due respect, we are not oversimplifying: we are calling a spade a spade.**

L41: While some of the earliest model studies used very small ensemble sizes, most past studies used ensembles of some reasonable number and quantified the statistical significance of the ensemble mean response, taking into account natural variability. It is therefore unjustified to claim that "much of the earlier literature had failed to properly account for the large internal variability associated with the stratospheric polar vortex and with the NAO".

**Again, we politely disagree with the referee. Yes, many recent studies have analyzed ensembles of model runs. But most used them incorrectly. Nearly all papers we are aware of compared to the ENSEMBLE MEAN of their model (or models) to the observations. As they found no warming in the model MEAN, they concluded the models were unable to reproduce the observations. The key point that was missed is that, for each eruption, the observations are just one realization of the system, and thus should not be compared to the mean of many model runs. We invite the referee to reread Driscoll et al (2012), who analyzed the CMIP5 models, just to cite one example: in their conclusion they state**

  **"None of the models simulate a sufficiently strong reduction in the geopotential height at high latitudes, and correspondingly the MSLP pressure fields and temperature fields show major differences with respect to the observed anomalies. This is despite some models having 10 ensemble members, giving a potentially strong signal-to-noise ratio."**

**See what they're saying? "We have averaged as many as 10 members, and yet the surface temperature in the models does not look like the observations: ergo, the models must be wrong." We believe this is a methodological flaw that afflicts most of the extant literature on this subject, in addition to an uncritical acceptance of the observational claims.**

L164: The number of deaths caused the Krakatau eruption has absolutely no bearing on the expected relative magnitude of the winter warming signal, as the number of deaths depends strongly on the population living in proximity to the volcano.

**Of course, we agree with the referee. But the reason for citing that number here is simply to remind the reader how destructive that eruption was. Krakatau is universally described as a truly cataclysmic event. And yet, as we show in our paper, the OBSERVED warming anomalies over Eurasia in the following winter were TOTALLY UNREMARKABLE, because the internal variability of the extratropical atmospheric circulation is very large, and an events as large as the 1883 Krakatau eruption is unable to overcome it. This is the key point of our work: that internal variability is MUCH LARGER than the impact eruptions as big as Krakatau (and Pinatubo).**

L224: The response of the two most extreme ensemble members illustrates that there
is natural variability, and that the anomaly in any one post-volcanic year may vary from
case-to-case, but it does not negate the possibility of a non-zero mean response, i.e.,
a higher probability of either negative or positive anomaly.

**We are agreed.  The large spread between the two extreme members "does not negate the
possibility of a non-zero mean response".  However, all the studies based on CMIP3 and
CMIP5 models report PRECISELY a zero mean response, including the 100-member ensemble
analyzed in our paper for Krakatau.  Showing the two extremes is the best way to
illustrate just how LARGE that variability is, and thus how tiny the forced response is,
a fact that had not been appreciated until very recently.  Hence our emphasis.**

L231: "acceleration"

**Thanks for flagging this typo.  We have corrected it.**

L257: Wunderlich and Mitchell didn't look at winter warming in the CMIP5 models.

**Yes, they looked only the NAO, which they found NOT to be anomalous after large volcanic
eruptions, in agreement with our results and many other papers: BUT the whole reason for
their study was to understand the lack of winter warming in those models.  So, that
citation is entirely appropriate in context.**

L258: This misrepresents the results of Stenchikov et al. (2006) who state in their
abstract "The IPCC models tend to simulate a positive phase of the Arctic Oscillation in
response to volcanic forcing similar to that typically observed. However, the associated
dynamic perturbations and winter surface warming over Northern Europe and Asia in
the post-volcano winters is much weaker in the models than in observations."

**Well... the claims in the abstract of Stenchikov et al (2006) regarding the CMIP3
models are, we fear, not quite representative of what one reads in the paper itself.**

**First, they only analyze 7 models, most of which are low-top, and hence with very poor
stratospheric resolution.  Second: two of the nine eruptions averaged in that study
(Tarawera and Bandai) are not in the tropics which, again, confuses things.  Third,
they average both the the first and second winters, which confuses things even more;
also, for 2 eruptions the winters were shifted by 1 year as the eruptions occurred in
October (we find this indefensible, and adds even more confusion).  Fourth: only 3 of
the 7 models (i.e. less than half!) show any statistically significant warming over
Eurasia at all (see their Figure 2), and in all cases it is much smaller than the
observed one.  So, this is pretty thin evidence for in support of the winter warming
theory via the stratospheric pathway, using models which don't even simulate the
stratosphere.**

**But there's more.  The follow-up study with the CMIP5 models (Driscoll et al, 2012),
explicitly says that the models in Stenchikov et al (2006) did a poor job in simulating
the winter warming response to volcanic eruptions, and that the CMIP5 models, alas, are
no better!  We quote from the Driscoll et al study (S06 is Stenchikov et al, 2006):**

 **"With substantially different dynamics between the models it was hoped to find at
  least one model simulation that was dynamically consistent with observations, showing
  improvement since S06.  Disappointingly, we found that again, as in S06 , despite
  relatively consistent post volcanic radiative changes, NONE OF THE MODELS MANAGE TO
  SIMULATE A SUFFICIENTLY STRONG DYNAMICAL RESPONSE."  (emphasis ours)**

**So there you have it: the authors are "disappointed" that NONE of the CMIP5 models
agrees with observations, and state that these models are no better than the CMIP3
models analyzed by Stenchikov et al (2006).  We agree with them.**

Footnote 4: For the sake of balanced consideration of prior work, reference to Zambri
and Robock (2016) should come in the introduction rather than here near the end of
the paper. Also, editorial commentary characterizing the work as "a single dissenting
voice" or "not, to date, ... independently reproduced" is clearly rhetoric meant to under-
mine confidence in this study, and would benefit from being recast in more objective
and quantitative terms.

**Why should we mislead the readers early on in our paper, by giving the false impression
that there are two equally-weighted sides to the story, when in fact ALL the studies which
analyzed the CMIP5 models are unanimous in reporting the lack of a winter warming, EXCEPT
for that one paper?  The Zambri and Robock (2016) paper is an outlier, and no other study
to date has independently backed their claims.  This is an objective statement, is it not?**

**A FINAL NOTE OF REASSURANCE: We sincerely hope Dr Toohey will not be taken aback by the
strong tone of our response to his comments.  We hold him in great respect, and are
very grateful for his time and consideration.  While we strongly disagree on the validity
of the winter warming theory, we look forward to his reply and will carefully consider it.**

---

## Author Response (AR2)

New York City
03 October 2020

Dear Dr. Palm:

Thank you for noting a couple of typos in the revised version: we apologize
for the oversight.  We have correctd them and, we believe, the manuscript
is now ready for publication.

It has been a real pleasure dealing with you.

With our very best regards,

Lorenzo Polvani and Suzana Camargo
Columbia University